# Knowledge, attitudes and practice towards yellow fever among nomadic populations: A cross-sectional study in yellow fever outbreak communities in Ghana

**Abdul-Wahab Inusah** [1], **Gbeti Collins**[2], **Peter Dzomeku**[3], **Michael Head** [4]*, **Shamsu-Deen Ziblim**[5]

1 Department of Global and International Health, School of Public Health, University for Development Studies, Tamale, Ghana, 2 Department of Behavioural Sciences, School of Public Health, University for Development Studies, Tamale, Ghana, 3 Directorate of Academic Planning & Quality Assurance (DAPQA) University for Development Studies, Tamale, Ghana, 4 Clinical Informatics Research Unit, Faculty of Medicine, University of Southampton, Southampton, United Kingdom, 5 Department of Internal Medicine, Tamale Teaching Hospital, Tamale, Ghana

\* m.head@soton.ac.uk

**Data Availability Statement:** All relevant data are within the paper and its Supporting Information files.

## Abstract

Despite the government and global health initiatives toward yellow fever epidemic control in Ghana, the country continues to witness sporadic outbreaks of yellow fever mostly among the unvaccinated population and suspected migrates(nomadic) who enter the country. Little is known about nomadic knowledge, attitudes and practice regarding this communicable disease in Ghana. We conducted a community-based cross-sectional survey in 22 yellow fever outbreak communities to assess nomadic household heads' knowledge, attitudes and practices (KAP) regarding yellow fever after an outbreak in November 2021 outbreak. Our study results were analyzed using descriptive statistics, bivariate and multivariate logistics regression with dichotomous outcomes. Significant statistics were obtained from multivariate analysis. About 90% of the nomadic had poor knowledge of the signs and symptoms of yellow with only 16% knowing the vector that transmits yellow fever. The most common source of information on yellow fever was the health campaign. Over 80% of household heads surveyed had positive attitudes regarding yellow fever with about 84% worried about the disease outbreak in their community. In a multivariate logistic regression model, age group(AOR = 2.79; 95% CI: 1.31, 5.98, p = 0.008)., gender ideology(AOR = 2.27; 95% CI: 1.14–4.51, p = 0.019), occupation(AOR = 15.65; 95% CI: 7.02, 34.87, p<0.001), source of health information(AOR = 0.27; 95% CI: 0.07, 0.96, p = 0.043), duration of stay in the community(AOR = 1.11; 95% CI: 1.31, 5.98, p = 0.008) and nationality (AOR = 0.22; 95% CI:0.47, 0.47, p<0.001) were associated with positive attitudes towards yellow fever. Close to 74% have a positive practice, with 97.3% controlling mosquitoes in their household. Nationality (AOR = 3.85; 95% CI: 2.26, 6.56, p<0.001), duration of stay in the community (AOR = 1.06; 95% CI: 1.03, 1.10, p = 0.001), and age group(AOR = 040; 95% CI: 0.22, 0.73, p = 0.003) were associated with positive practices. Our findings show that yellow fever KAP was variable with clear knowledge gaps. Regular locally-tailored education and health

**Funding:** The authors received no specific funding for this work.

**Competing interests:** The authors have declared that no competing interests exist.

promotion campaigns should be considered to improve knowledge and preventive practices against this infectious disease.

## Introduction

Yellow fever is identified by the WHO as a Neglected Tropical Disease [1]. It is a viral vector-borne disease that affects 47 countries in tropical zones, especially in South America, Central America and 34 sub-Sahara African countries [2, 3]. There are an estimated 29,000–60,000 deaths globally each year [4]. The yellow fever virus is transmitted by the bite of *Aedes* or *Haemagogus* mosquito species [5]. The most common symptoms include fever, jaundice, body pain, abdominal pain, vomiting, and gum haemorrhage. Where yellow fever is endemic, countries often require proof of vaccination upon arrival from incoming travellers [2, 6].

In 2020 the yellow fever burden on the African continent has risen with Ghana recording the highest incidence of 1267 cases per 100,000, with 10,350 confirmed cases [3]. The case fatality rate in Ghana is estimated at 10%-17%.

To date, there is no known treatment for yellow fever; therefore, prevention through vaccination is noted to be essential in avoiding the risk of associated morbidity and mortality [6, 7].

Cases of yellow fever in Ghana are mostly reported among nomadic populations migrating into the country. The Upper West and Savannah regions contain forest reserves often serving as tourist sites where nomadic migrants find work [7].

There will be significant under-reporting, yellow fever infections may not be detected due to inadequate surveillance and reporting for vaccination and other interventions especially among this population [8]. Inadequate knowledge, negative attitudes and poor practices of the different populations on travel health and more specifically yellow fever contribute largely to the incidence of yellow fever [7, 9].

A multi-country comparison found African participants to typically have more knowledge of yellow fever compared to those from the East Mediterranean and other countries, with males and older age groups having highest knowledge [10]. More than three-quarters of travellers were unaware of yellow fever infection and vaccination in India [11]. Other studies highlighted how knowledge was found to be associated with the geographical location of birth and fields of academic study [12], although a community study conducted in Southern Ethiopia found low population knowledge on transmission modes, cause and preventive strategies.

Yellow fever control in Ghana is based upon tested of suspected cases, field investigations to determine likely vaccination status of exposed individuals, with vaccination campaigns sometimes deemed necessary to bring any outbreak under control [13]. There is very little known about nomadic knowledge, attitude and practices regarding yellow fever in West Africa. This study seeks to assess the knowledge, attitude, and practices among nomadic populations in the Savannah region of Ghana.

## Methods

### Study area, design and period

A community-based cross-sectional survey was conducted between February to March 2022 among nomadic households in the 22 yellow fever outbreak communities in the West Gonja Municipal of the Savanna Region of Ghana (Fig 1). The Municipal is one of the seven districts of the Savanna region which also serves as the administrative capital of the region. It is close to the borders of the Ivory Coast and Burkina Faso. The Municipal has a landmass of

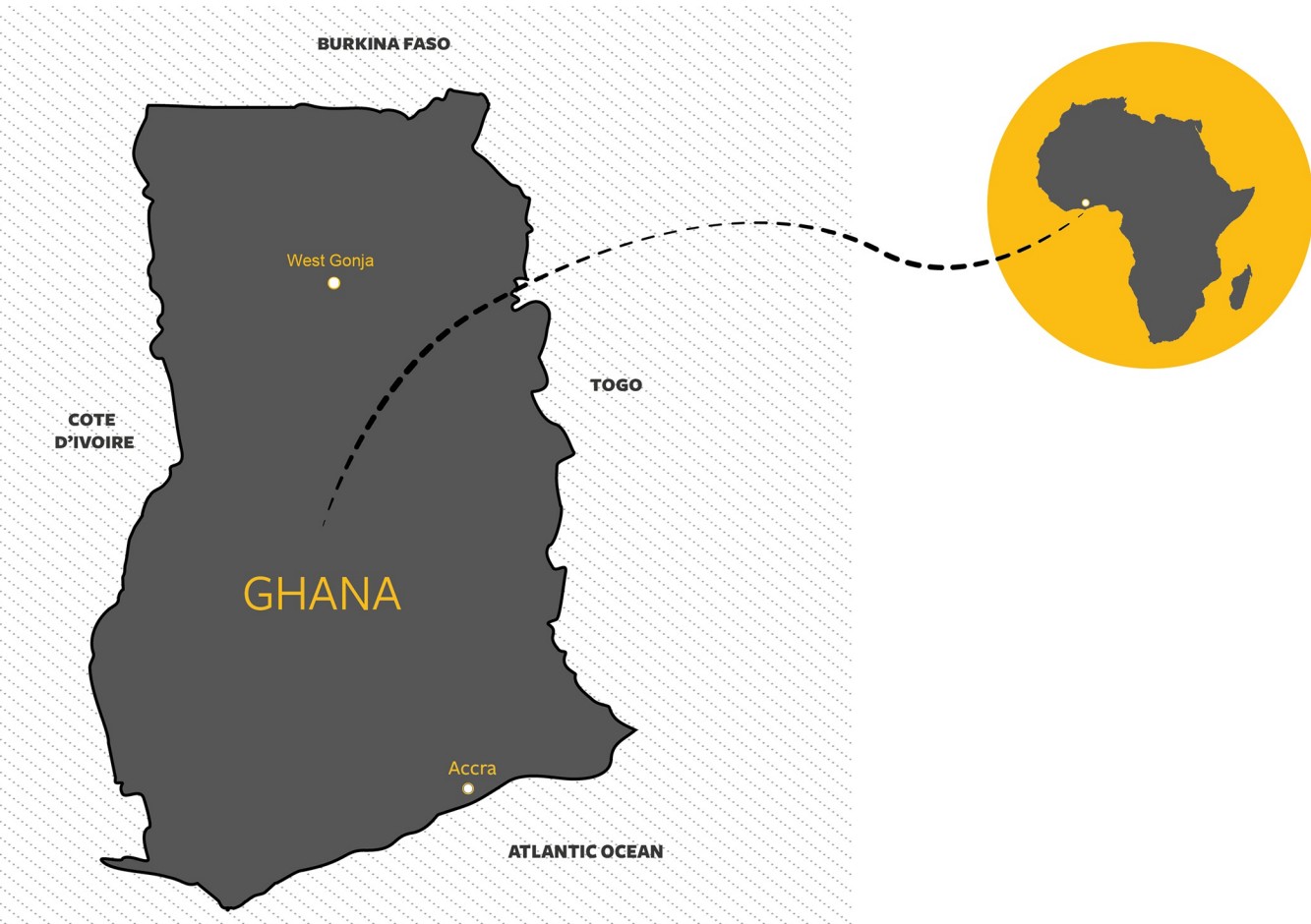

**Fig 1. Map identifying the study location, West Gonja Municipal in Savanna Region of Ghana, West Africa.** Image drawn within our research group, the Clinical Informatics Research Unit at University of Southampton.

4715.9sqkm, part of which is occupied by a protected forest reserve known as the Mole National Park [14]. The Savannah region is humid, with the forest and accompanying water sources providing a breeding ground for mosquitoes. Transmission of malaria also high in the area throughout the year [15].

### Study population, participants, and inclusion criteria

The study population was all nomadic households in yellow fever affected communities at the time of the data collection. Household heads or spouses who were present in their households and consented were the study participants. The term 'nomadic' here refers to a group of people who wander around in search of pasture to feed their livestock, fertile land or both.

Using the yellow fever line list *(a table containing detailed information on each case of a disease outbreak)* obtained from the Municipal Assembly Health Directorate, all communities with confirmed yellow fever cases were purposively selected for the study. In each community, a community health volunteer (CHV), who previously supported the most recent yellow vaccination campaign, was used as a focal person to purposively identify all nomadic households for the research officers. Using snowballing approaches, consented household heads/spouses were also asked to identify other nomadic households who may meet the inclusion criteria.

### Sample size, sampling technique

Required sample size was 403, at a 95% confidence level. We assumed the prevalence of positive knowledge, attitudes, and practices toward yellow fever to be 50%, margin of error of 5% and inflated the sample size by 5% for non-response and incomplete data entry.

### Data collection tool, and procedure

A structured questionnaire was adapted from similar studies [11, 16–18]. The study tool had four (4) distinct sections; **Section (1)** obtained demographic information including included gender, age (years), marital status, religious affiliation, household size, nationality, and educational level. **Section (2)** examine respondents' knowledge of yellow fever (YF) including symptoms and transmission modes. **Section (3)** collected data on respondents' attitudes towards yellow fever. **Section (4)** covers the various preventative practices adopted by the households and Sources of information regarding yellow fever.

The questionnaire was uploaded onto the Android smartphone App. (ODK) and pretested in the North Gonja District among similar study subjects. A face-to-face interview technique was used by trained research officers. This approach was adopted because it is thought that most of the study population is unable to read. Therefore, the questions were read out in local dialects, predominantly Hausa, Dagbani, Fulani and Gonja.

### Data analysis

Data was exported into Microsoft Excel 2019 for cleaning. The data analysis was carried out using Stata version 15. Data were analysed using descriptive and inferential statistics.

Key dependent variables were knowledge, attitudes and practices towards yellow fever. To assess the outcome variables, a score of one (1) was given a correct response, whilst zero (0) was assigned to the incorrect/ Don't know response. This was based on the WHO yellow fever protocol [19].

Knowledge—To summarize overall knowledge of yellow fever, 24 items were used(two items have more than 1 correct answer) and all those whose total scores were above 13 out of 24 were considered as having "good knowledge", whilst those with total scores less than 13 were considered "poor knowledge". This scoring system is adapted from a previous similar study [12]. The highest knowledge score in this study was 11, with only 3 participants scoring 11 out of 24. This made the knowledge data not suitable for regression analysis.

Attitudes—To assess attitudes towards yellow fever, 12 variables were used. To assess the overall attitudes, a 60% cut-off point was used. All those scores above the 60%(7.2 and above) were considered as having "positive attitudes" and all those with total scores less than 60% were considered "negative attitudes" [16].

Practice—Various practices adopted by households toward yellow fever were assessed using 9 variables. As previously studied [18], 55% was used as the cut-off point(5 and above out-off 9). Again, all those scoring 5 and above, were considered as having "good practice", and those with a total score less than 5 were considered "poor practice".

### Key independent variables

The independent variables were all categorical, these included, the age group, gender, marital status, religion, family size, Nationality, Occupation, and duration of stay in current community.

Bivariate and multivariate logistics regression was conducted to assess the impact of key independent variables on the dependent variables. The strength of association between the

predicting and outcomes variables was determined using the adjusted odds ratio. To establish statistical evidence of the relationship, p-values less than 0.05 were considered statistically significant. The model fitness was checked using Hosmer-Lemeshow Positiveness-of-fit test (p-value> 0.05 considered no evidence of poor fitness).

### Ethical consideration

Ethical approval for the study was obtained from the University for Development Studies (UDS) Research and Ethics Review Board. Also, permission was sought from the Savanna Regional Health Directorate through an introductory letter. The purpose of the study was explained to the study subjects and participants gave their informed written consent. At the end of each interview, research officers spent further time educating the household on the signs and symptoms of yellow fever and various prevention strategies.

## Results

A total of 414 households participated in the survey (Table 1). All household heads invited to participate in the study accepted our invitation. Among the study participants, 57.7% were males, with a mean age of 38.5± 13.1, and a full age range of 18 to 84 years. By relationship status, 91% of the participants were married, and the main occupation among participants was Herdsman (67.4%). By nationality, 56% of the participants were foreign nomadic, and the majority migrated from the Benin republic. All 414 households (100%) reported their religion as Islam.

### Knowledge of the signs and symptoms, mode of transmission and sources of information on yellow fever

In this study, only those who have heard of yellow fever were included the study. Out of the 414 participants, 73.7% received information on yellow fever through health campaigns, with only 0.7% receiving information on yellow fever information through the Media (Fig 2). A majority (92.8%) of the participants do not know what causes yellow fever, with only 4.6% who rightly said yellow fever is caused by a virus. When participants were asked what transmits yellow fever, only 16.2% rightly said a bite of an infected mosquito, and 47% believed yellow fever can be transmitted from person to person. Also, only 38.4% believed infected monkeys can transmit yellow fever to a person. A majority of the participants (56.3%) don't know the highest-risk time the Aedes mosquitoes are likely to bite. When participants were asked about signs and symptoms, 27.5% mentioned fever, 15.7% said vomiting of blood, and 7.7% mentioned headache. However, 55% did not know the main sign and symptoms of yellow fever. Stagnant waters (27.5%), water containers (8.5%) and septic tanks (7.7%) respectively were cited as the main breeding sites for the yellow fever vector. The maximum knowledge score was 11 out of 24 questions, mean score of 4.2 ± 2.7, thus indicating all participants have poor knowledge of yellow fever. Table 2 presents data on participants' knowledge of yellow fever.

### Nomadic attitudes towards yellow fever

Table 3 shows participants' attitudes towards yellow fever. From the study, a majority (65.5%) of the participants have heard of the yellow fever outbreak in their district. Closed to 77.2% of participants believed yellow fever was a serious illness, with 83.6% expressing worry about the yellow fever outbreak. A high proportion (71.5%) of the participants believed that the disease affects all age groups, and 87.2% believed that their families were at risk of yellow fever infection. More than 86% of the participants believed that unvaccinated people were at higher risk

**Table 1. Sociodemographic characteristics of nomadic in the West Gonja Municipal Ghana, 2022.**

| Variable | Frequency N = 414 | Percentage% |
|---|---|---|
| **Age group** | **Mean 38.5 ± 13.1** | |
| 18–34 | 187 | 45.2 |
| 35–51 | 152 | 36.7 |
| 52–68 | 64 | 15.5 |
| 69–85 | 11 | 2.7 |
| **Household Size (number of people)** | **Mean 7.04 ± 3.7** | |
| 1/5 | 164 | 39.6 |
| 6/10 | 189 | 45.7 |
| 11$^+$ | 61 | 14.7 |
| **Gender** | | |
| Female | 175 | 42.3 |
| Male | 239 | 57.7 |
| **Marital status** | | |
| Never married | 21 | 5.1 |
| Married | 375 | 90.6 |
| Widowed | 18 | 4.4 |
| **Main occupation** | | |
| Agro-pastoralist | 114 | 27.5 |
| Herdsman | 279 | 67.4 |
| Trader/Vendor | 21 | 5.1 |
| **Nationality** | | |
| Ghanaian | 184 | 44.4 |
| Foreigner | 230 | 55.6 |
| **Foreign country** | | |
| Benin | 119 | 51.7 |
| Burkina Faso | 36 | 15.7 |
| Nigeria | 55 | 23.9 |
| Togo | 20 | 8.7 |
| **Relocated in last one year** | | |
| No | 368 | 88.9 |
| Yes | 46 | 11.1 |
| **Plan to relocate within six months** | | |
| No | 365 | 88.2 |
| Undecided | 36 | 8.7 |
| Yes | 13 | 3.1 |

of yellow fever. About 91% believed the yellow fever vaccine is safe for their families, whilst 91.6% trusted the government's ability to respond to the yellow fever outbreak.

## Overall attitudes toward yellow fever

Table 4 shows the overall practices towards yellow fever. From the results, a majority (80.2%) of the participants had positive attitudes towards yellow fever. Participants aged 35–51 were 2.8 times more likely to have positive attitudes toward yellow fever compared to 18–34 years. (AOR = 2.79; 95% CI: 1.31, 5.98, p = 0.008). Being male participants was significantly associated with having positive attitudes towards yellow fever; the odds of having positive attitudes toward yellow fever were 2.27 higher in men compared to females. (AOR = 2.27; 95% CI:

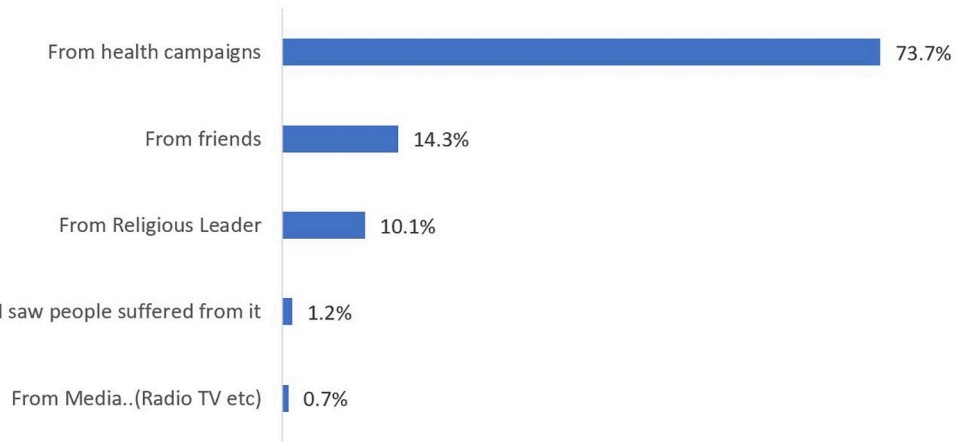

**Fig 2. Main source of YF information.**

1.14–4.51, p = 0.019). Similarly, being a herdsman, and trader/vendor was significantly associated with Positive attitudes towards fever compared to being Agro-pastoralist (AOR = 15.65; 95% CI: 7.02, 34.87, p<0.001), (AOR = 6.21; 95% CI: 1.54, 25.08, p = 0.010) respectively. Foreign nomadic were 0.22 times less likely to have Positive attitudes toward yellow fever compared to native nomadic. (AOR = 0.22; 95% CI:0.47, 0.47, p<0.001). Those who have relocated within the last year were 10.55 times more likely to have Positive attitudes than those who have not (AOR = 10.55; 95% CI: 2.54, 43.89, p = 0.001). Duration(months) of stay was significantly associated with having Positive attitudes toward yellow fever (AOR = 1.11; 95% CI: 1.31, 5.98, p = 0.008). Also, receiving of information on yellow fever from health campaigns and media were significantly associated with Positive attitudes (AOR = 0.27; 95% CI: 0.07, 0.96, p = 0.043), (AOR = 0.01; 95% CI: 0.00, 0.003, p<0.001) respectively.

## Nomadic practices toward yellow fever

Table 5 shows various practices adopted by the study participants. Almost all participants (97.3%) had some strategies used to reduce mosquitoes in their households. More than 81% of the participants prevent standing water around their homes to reduce the breeding of mosquitoes, 62% used insecticide-treated net, 44% uses smoke to drive away mosquitoes, whilst 85% cover their bodies properly with clothes to avoid mosquito bites. Again, 92% of the participant properly cover water-holding containers, and 66% turned empty containers upside down to avoid the breeding of mosquitoes.

## Overall practices towards yellow fever

From the analysis, 72.7% of the participants had a Positive practice with yellow fever. Compared with younger ages, being within the age group of 35 to 51 and 52 to 68 were significantly associated with having poor practices on yellow fever (AOR = 040; 95% CI: 0.22, 0.73, p = 0.003) and (AOR = 0.34; 95% CI: 0.15, 0.77, p = 0.010) respectively (Table 6). Foreign nomadic were significantly associated with having Positive practices on yellow fever (AOR = 3.85; 95% CI: 2.26, 6.56, p<0.001). Also, the longer you stay in your current location, the more knowledgeable you were; an additional a month's stay was significantly associated with Positive practices on yellow fever. (AOR = 1.06; 95% CI: 1.03, 1.10, p = 0.001). Table 7 illustrates a summary of the findings associated with the outcome variables.

**Table 2.  Knowledge of yellow fever, among nomadic populations in the West Gonja Municipal Ghana, 2022.**

| Variable | Frequency N = 414 | Percentage% |
|---|---|---|
| **Cause of Yellow fever** | | |
| Don't know | 384 | 92.8 |
| Virus | 19 | 4.6 |
| Bacteria | 8 | 1.9 |
| Germs | 3 | 0.7 |
| **YF transmission vector** | | |
| Don't know | 318 | 76.8 |
| Bite of an infected mosquito | 67 | 16.2 |
| Bite of an infested tsetse fly | 18 | 4.4 |
| Bite of an infested housefly | 11 | 2.7 |
| **The same mosquito transmits malaria** | | |
| Don't know | 16 | 23.9 |
| Yes | 48 | 71.6 |
| No | 3 | 4.5 |
| **Patient to person transmit YF** | | |
| Don't know | 194 | 46.9 |
| Yes | 194 | 46.9 |
| No | 26 | 6.3 |
| **Can a monkey transmit YF to person** | | |
| Don't know | 218 | 52.7 |
| Yes | 159 | 38.4 |
| No | 37 | 8.9 |
| **YF transmitted from food and water** | | |
| Don't know | 219 | 52.9 |
| Yes | 149 | 36.0 |
| No | 46 | 11.1 |
| **Time YF vector most likely to feed/bite** | | |
| Don't know | 233 | 56.3 |
| Day | 34 | 8.2 |
| Night | 34 | 8.2 |
| Both | 113 | 27.3 |
| **Can the YF vector breed inside homes** | | |
| Don't know | 191 | 46.1 |
| Yes | 192 | 46.4 |
| No | 31 | 7.5 |
| **Covering/removal of stagnant water prevents the breeding of YF vector** | | |
| Don't know | 170 | 41.1 |
| Yes | 226 | 54.6 |
| No | 18 | 4.4 |
| **The pouring of chemicals into stagnant water kills the YF larvae** | | |
| Don't know | 180 | 43.5 |
| Yes | 213 | 51.5 |
| No | 21 | 5.1 |
| **Signs and Symptoms of YF include*** | | |
| Don't know/incorrect response | 225 | 54.4 |
| Fever | 114 | 27.5 |
| Headache | 32 | 7.7 |

(*Continued*)

**Table 2.** (Continued)

| Variable | Frequency N = 414 | Percentage% |
|---|---|---|
| Jaundice | 33 | 8.5 |
| Muscle/body pains | 13 | 3.1 |
| Skin rashes | 1 | 0.2 |
| Bleeding | 1 | 0.2 |
| Abdominal pain | 5 | 1.2 |
| Vomiting blood | 65 | 15.7 |
| **Breeding Site for yellow fever vector includes**[*] | | |
| Don't know/incorrect response | 312 | 54.4 |
| Stagnant water | 83 | 27.5 |
| Septic tanks | 2 | 7.7 |
| Water containers | 9 | 8.5 |
| False bananas | 19 | 3.1 |
| Drains and garbage | 32 | 0.2 |
| **Overall knowledge of YF** | **Mean 4.2 ± 2.7** | |

Questions with Asterisk[*] indicate multiple responses and where multiple right answers

## Discussion

The study investigated nomadic knowledge, attitude and practices regarding yellow fever in Ghana. The study involved only nomadic who were aware of yellow fever. We found overall knowledge to be low or poor.

Our findings are similar to previous research in different populations, [12] where an Ethiopian study found in which only 9.6% of the university student participants were found to have adequate overall knowledge of yellow fever. These results give cause for concern, as the nomadic population are most likely not to have put in place adequate preventive measures. Findings here demonstrated low knowledge on the cause of yellow fever, main transmission vector, breeding point for the transmission vector, and hygienic management of breeding sites.

The low knowledge of these indicators in our study is similar to previous research from India and Ethiopia [12, 16]. Other studies [11, 16] found very few participants knew that the mosquito is the transmission vector for yellow fever. A Vietnam study about KAP of dengue fever included participants who were hospitalized with a fever; findings highlighted a low average knowledge score about dengue, with recommendations including the need for increased health promotion through schools [20]. Similar findings have been observed for malaria [21].

However, our study did reveal areas of widespread knowledge about yellow fever. with health campaigns being were the major source of information. Ghanaian yellow fever vaccination campaigns are typically accompanied by extensive education and sensitization by health workers in. Different populations receive health information about mosquito-borne disease via other sources, for example from TV messaging [17, 22].

When considering attitudes, we found high (65.5%) awareness of the recent yellow fever outbreak among the nomadic population. Reported awareness has been high in Ethiopian studies, reporting 83% and 86% awareness [12, 23], but only 25% awareness from an Indian study [11], It will therefore be good for Ghanaian health stakeholders to take advantage of the high awareness among the nomadic population to scale up interventions to help eradicate or reduce the incidence of YF.

**Table 3. Attitudes of nomadic populations in yellow fever outbreak communities in the West Gonja Municipal, Ghana, 2022.**

| Variable | Frequency N = 414 | Percentage% |
|---|---|---|
| **Heard of the YF outbreak in this district** | | |
| Yes | 271 | 65.5 |
| No | 121 | 29.2 |
| I am not sure | 22 | 5.3 |
| **Yellow fever is a serious illness and I am careful about it** | | |
| Yes | 319 | 77.1 |
| No | 3 | 0.7 |
| I am not sure | 92 | 22.2 |
| **I am worried about the YF outbreak in and around your Municipal** | | |
| Yes | 346 | 83.6 |
| No | 10 | 2.4 |
| I am not sure | 58 | 14.0 |
| **YF affects all age groups and I can be infected** | | |
| Yes | 296 | 71.5 |
| No | 7 | 1.7 |
| I am not sure | 111 | 26.8 |
| **YF is a fatal/killer disease and I killed if infected** | | |
| Yes | 306 | 73.9 |
| No | 2 | 0.5 |
| I am not sure | 106 | 25.6 |
| **Fear that you /your family are at risk for YF** | | |
| Yes | 361 | 87.2 |
| No | 4 | 1.0 |
| I am not sure | 49 | 11.8 |
| **Unvaccinated persons are at risk of YF, I will make myself available for vaccination** | | |
| Yes | 359 | 86.7 |
| No | 2 | 0.5 |
| I am not sure | 53 | 12.8 |
| **YF is an easily treatable disease and I will go for the treatment if I am infected** | | |
| Yes | 129 | 31.2 |
| No | 130 | 31.4 |
| I am not sure | 155 | 37.4 |
| **A person who is working/living in a forest area is at high risk of getting YF. Are you worried about this?** | | |
| Yes | 284 | 68.6 |
| No | 4 | 1.0 |
| I am not sure | 126 | 30.4 |
| **Are you scared of being infected with YF?** | | |
| Yes | 370 | 89.37 |
| No | 8 | 1.93 |
| I am not sure | 36 | 8.7 |
| **Do you think the YF vaccine is safe for you/family?** | | |
| Yes | 374 | 90.6 |
| No | 2 | 0.5 |
| I am not sure | 37 | 9.0 |

(*Continued*)

**Table 3.** (Continued)

| Variable | Frequency N = 414 | Percentage% |
|---|---|---|
| **Do you trust the Ghanaian government's response to the YF outbreak?** | | |
| Yes | 379 | 91.6 |
| No | 4 | 1.0 |
| I am not sure | 31 | 7.5 |
| **Overall Attitudes towards YF** | | |
| Negative attitudes | **82** | **19.8** |
| Positive attitudes | **332** | **80.2** |

**Table 4. Bivariate and multivariate logistic regression analyses showing predictors of attitudes towards yellow fever in the West Gonja Municipal, Ghana, 2022.**

| Independent Variable | OR | (95%Cl) | P-value | aOR | 95%Cl | P-value |
|---|---|---|---|---|---|---|
| **Age group** | | | | | | |
| 18–34 | 1 | | | 1 | | |
| 35–51 | 1.64 | 0.95–2.85 | 0.078 | 2.79 | 1.31–5.98 | 0.008** |
| 52–68 | 1.21 | 0.60–2.42 | 0.596 | 1.71 | 0.61–4.80 | 0.307 |
| 69–85 | 3.08 | 0.38–24.71 | 0.290 | 3.79 | 0.36–40.28 | 0.270 |
| **Gender** | | | | | | |
| Female | 1 | | | 1 | | |
| Male | 2.14 | 1.31–3.49 | 0.002 | 2.27 | 1.14–4.51 | 0.019** |
| **Marital status** | | | | | | |
| Never married | 1 | | | 1 | | |
| Married | 0.42 | 0.11–1.85 | 0.252 | 0.47 | 0.09–2.36 | 0.359 |
| Widowed | 0.27 | 0.05–1.63 | 0.115 | 0.35 | 0.04–3.03 | 0.336 |
| **Main occupation** | | | | | | |
| Agro-pastoralist | 1 | | | 1 | | |
| Herdsman | 6.52 | 3.80–11.18 | <0.001 | 15.65 | 7.02–34.87 | <0.001*** |
| Trader/Vendor | 1.82 | 0.66–5.03 | 0.249 | 6.21 | 1.54–25.08 | 0.010** |
| **Nationality** | | | | | | |
| Ghanaian | 1.00 | | | 1 | | |
| Foreigner | 0.41 | 0.24–0.70 | <0.001 | 0.22 | 0.10–0 .47 | <0.001*** |
| **Duration of stay** | 1 | | | 1 | | |
| | 1.09 | 1.05–1.14 | <0.001 | 1.11 | 1.04–1.18 | 0.001** |
| **Relocated in last one year** | | | | | | |
| No | 1 | | | 1 | | |
| Yes | 3.92 | 1.18–12.96 | 0.025 | 10.55 | 2.54–43.89 | 0.001** |
| **Source of information** | | | | | | |
| From friends | 1 | | | 1 | | |
| From health | 0.16 | 0.05–0.52 | 0.002 | 0.27 | 0.07–0.96 | 0.043* |
| campaigns | 0.12 | 0.01–1.54 | 0.101 | 0.01 | 0.00–0 .03 | <0.001*** |
| From Media..(Radio T..) | 2.20 | 0.22 21.89 | 0.502 | 3.91 | 0.30–51.51 | 0.299 |
| | ———— | ———————— | | ——————— | —————— | ——— |
| From Religious Leader I saw people suffer from it | | | | | | |

Note:OR, odds ratio

aOR, adjusted odds ratio

CI, confidence intervals

P-value<0.05, Hosmer-Lemeshow chi2(8) = 8.88. Prob > chi2 = 0.3524.

**Table 5. Nomadic practices toward yellow fever in yellow fever outbreak communities in the West Gonja Municipal, Ghana, 2022.**

| Variable | Frequency N = 414 | Percentage% |
|---|---|---|
| **Do you do anything to reduce mosquitoes?** | | |
| Yes | 403 | 97.3 |
| No | 11 | 2.7 |
| **Do you prevent standing water around the house to reduce mosquitoes?** | | |
| Yes | 328 | 81.4 |
| No | 75 | 18.6 |
| **Do you use insecticide-treated nets to protect against mosquitoes in the home?** | | |
| Yes | 249 | 61.7 |
| No | 154 | 38.2 |
| **Do you use smoke to drive mosquitoes away?** | | |
| Yes | 177 | 43.9 |
| No | 226 | 56.1 |
| **Do you cover your body with clothes to protect against mosquitoes?** | | |
| Yes | 342 | 84.9 |
| No | 61 | 15.1 |
| **Has the government come to spray insecticide to reduce mosquitoes?** | | |
| Yes | 13 | 3.1 |
| No | 401 | 96.9 |
| **Do you cover water containers in the home?** | | |
| Yes | 380 | 91.8 |
| No | 34 | 8.21 |
| **Do you clean water filed containers and ditches around the house?** | | |
| Yes | 357 | 86.2 |
| No | 57 | 13.8 |
| **Do you turn containers upside down to avoid water collection?** | | |
| Yes | 273 | 65.9 |
| No | 141 | 34.1 |
| **Overall practice towards YF** | | |
| Poor Practice | 113 | 27.3 |
| Positive Practice | 301 | 72.7 |

Yellow fever vaccination campaigns are typically well-received in Ghana. A 2020 campaign in the Savannah region showed high trust in the health services and confidence in the vaccination resulting in a reported vaccine uptake of 99%, [24].

The fear of being at risk for yellow fever and someone working or living in the forest zones being at higher risk was again mentioned by a majority of the participants. Our findings in these regards were in line with a similar study result reported in Jinka, Ethiopia [12] and among the general population in South Omo Zone [16]. These positive attitudes of the nomadic population towards yellow fever put them in a position to most likely adopt preventive measures.

The current study results also show that most of the participants (68.8%) did not perceive or were not sure yellow fever was an easily treatable disease, adding to their fears of being infected by the disease. In tandem with the current study's results is that of [16] who reported a majority of participants indicated that yellow fever was not easily treatable and therefore feared they could easily be infected. As noted in this study's background, there is no known

**Table 6. Bivariate and multivariate logistic regression analyses showing predictors of practices towards yellow fever in the West Gonja Municipal, Ghana, 2022.**

| Independent Variable | OR | (95%Cl) | P-value | aOR | 95%Cl | P-value |
|---|---|---|---|---|---|---|
| **Age group** | | | | | | |
| 18–34 | 1 | | | 1 | | |
| 35–51 | 0.38 | 0.22–0 .62 | <0.001 | 0.40 | 0.22–0.73 | 0.003** |
| 52–68 | 0.38 | 0.20–0.72 | 0.003 | 0.34 | 0.15–0.77 | 0.010* |
| 69–85 | 0.96 | 0.21–4.67 | 0.964 | 0.71 | 0.12–3.99 | 0.683 |
| **Gender** | | | | | | |
| Female | 1 | | | 1 | | |
| Male | 0.54 | 0.34–0.86 | 0.009 | 0.63 | 0.36–1.07 | 0.091 |
| **Marital status** | | | | | | |
| Never married | 1 | | | 1 | | |
| Married | 0.43 | 0.12–1.48 | 0.18 | 0.58 | 0.14–2.47 | 0.465 |
| Widowed | 0.43 | 0.09–2.12 | 0.31 | 0.49 | 0.07–3.35 | 0.467 |
| **Main occupation** | | | | | | |
| Agro-pastoralist | 1 | | | 1 | | |
| Herdsman | 2.15 | 1.34–3.43 | 0.001 | 1.34 | 0.75–2.40 | 0.319 |
| Trader/Vendor | 3.91 | 1.09–14.06 | 0.037 | 1.49 | 0.35 6.25 | 0.587 |
| **Nationality** | | | | | | |
| Ghanaian | 1 | | | | 1 | |
| Foreigner | 2.81 | 1.79–4.40 | <0.001 | 3.85 | 2.26–6.56 | <0.001*** |
| **Duration of stay** | 1.02 | 0.99–1.05 | 0.121 | 1.06 | 1.03–1.10 | 0.001** |
| **Relocated in last one year** | | | | | | |
| No | 1 | | | | | |
| Yes | 0.60 | 0.32–1.14 | 0.122 | 0.62 | 0.29–1.34 | 0.225 |
| **Source of information** | | | | | | |
| From friends | 1 | | | | | |
| From health campaigns | 0.96 | 0.51–1.80 | 0.906 | 1.37 | 0.67–2.81 | 0.383 |
| From Media..(Radio T..) | | —— | | 1 | — | |
| From Religious Leader | '1 | 0.54–3.47 | 0.515 | 1.33 | 0.48–3.67 | 0.582 |
| I saw people suffer from it | 1.36 | 0.04–1.62 | 0.146 | 0.39 | 0.05–3.21 | 0.379 |
| | 0.25 | | | | | |

Note:OR, odds ratio

aOR, adjusted odds ratio

CI, confidence intervals

P-value<0.05, Hosmer-Lemeshow chi2(8) = 7.47. Prob > chi2 = 0.4866.

**Table 7. Summary of the study outcome variables.**

| Outcome variables | Overall score | |
|---|---|---|
| Knowledge | Mean, SD | 4.2 ± 2.7 |
| Attitude | Negative Attitude | 80.2 |
| | Positive Attitude | 19.8 |
| Practice | Poor Practice | 27.3 |
| | Positive Practice | 72.7 |

cure or treatment for yellow fever [1], except for the management of its signs and symptoms, making vaccination the surest way to avoid being infected. It was therefore not surprising that a significant proportion of our sampled respondents affirmed vaccines as a safe means of protection against the disease. Trust for the government's responses to the outbreak only affirms the position of the state in achieving 100% coverage of vaccination, education and sensitization on the disease. It is probably on this backbone that Ghana as one of the most endemic countries for yellow fever is equally leading vaccination coverage and herd immunity in Africa [25].

The overall attitude of the nomadic population as established by this current study was (80.3%). Our finding nevertheless disagrees with Endale et al., (2020) who found overall attitude (51.2%) among Jinka University students. We found foreign nomadic to be less likely to have Positive attitudes toward yellow fever compared to native nomadic. This is probably because native nomads are more stable and easily adjust to a positive attitude as compared to the foreign nomadic, who are usually highly mobile making attitudinal adjustment very difficult.

Our study found the overall practices of study participants to be very good. This result shows that the nomadic population was adopting preventive practices aimed at curbing the vector's breeding and spread of yellow fever. Our study found a majority of the participants mentioned that they kept measures to prevent the breeding of the yellow fever vector (mosquitoes). This includes the use of insecticide-treated nets, cleaning/draining stagnant water, and covering or turning upside down containers that can breed the vector. Some components of Itrat et al., (2008) study results in Karachi on preventive practices of dengue fever were in agreement with the current research findings except for the use of insecticide-treated nets which recorded a very low usage while mosquito sprays and coils were the most preferred [22]. However, mosquito nets, sprays or coils are well known effective preventive methods for the yellow fever vector.

Several studies have reported these methods to be the most effective means of prevention of yellow fever and related arboviral diseases [17, 26]. Water stagnation preventive measures to avoid breeding sites were also popular techniques respondents mentioned they adopted. This corroborates Itrat et al., (2008) findings in Karachi and [27] study in Thailand in which dengue vectors and associated hemorrhagic fever cases reduced significantly in areas where clean-up campaigns were organized before and during rainy seasons [27].

The use of smoke to drive away mosquitoes appeared not to be a popular choice for most of our study participants. The inconveniences associated with smoking during breathing could account for this result in our study. Again, we found government intervention in terms of mass insecticide spraying to be rare as indicated by the majority of the population we studied. In an era of the outbreak, it will be critical on the part of the government to for ones embark on mass fumigation of severely endemic zones to complement the routine efforts of the local populations. The absence of this implies that the nomadic population must bear all responsibility for preventing breeding and bites of the vector. This situation may impede the eradication goal desired nationally by Ghana and international targets set by the WHO.

Our findings found that foreign nomadic were about four times more likely to have Positive practices on yellow fever. With this, they are likely to put in place measures to prevent themselves from being infected by the disease or any other arboviral disease to remain healthy while visiting or working in the forest areas especially.

The COVID-19 pandemic has highlighted the need for decision-makers to better understand the behaviour and social dynamics of populations during outbreaks or other public health emergencies. This is particularly important with preventive measures such as vaccination campaigns, where understanding levels of confidence, reasons for hesitancy, and drivers of trust in public health messaging is vital information [28]. The viewpoints of rural or hard-

to-reach groups, where disease burden may be high and health programs such as mass drug administrations urgently require high uptake, should also be sought in order to best inform health promotion around the program rollout [29].

### Limitation of the study

Some of the limitations of this study include that this was a cross-sectional study design. Whilst the findings are statistically significant, this does not equate to clinical significance and thus limits our inference around causal association. The study was only conducted in yellow fever outbreak communities in the West Gonja Municipal, and may not be generalizable to other nomadic communities, to the wider Ghana population, or to other populations outside of Ghana. Local context may differ elsewhere.

## Conclusion

There are mixed findings around knowledge, attitude and practice towards yellow fever within nomadic populations in the Savannah region of Ghana. It is important to ensure that there is sustained health promotion among communities where there may be low existing vaccine uptake, raising awareness of yellow fever and the importance of immunization. Ghanaian approaches towards yellow fever control can be improved, including locally-tailored education and health promotion campaigns to improve awareness. Renewed evidence around yellow fever knowledge, attitude and practice in other Ghanaian and African populations would support evidence-informed decision-making for vaccination campaigns and public health messaging.

## Supporting information

**S1 File. Raw dataset.**
(DTA)

**S2 File. Study questionnaire.**
(DOCX)

## Author Contributions

**Conceptualization:** Abdul-Wahab Inusah, Gbeti Collins, Peter Dzomeku, Shamsu-Deen Ziblim.

**Data curation:** Abdul-Wahab Inusah, Gbeti Collins, Shamsu-Deen Ziblim.

**Formal analysis:** Abdul-Wahab Inusah.

**Methodology:** Abdul-Wahab Inusah, Gbeti Collins, Peter Dzomeku, Michael Head, Shamsu-Deen Ziblim.

**Project administration:** Abdul-Wahab Inusah.

**Validation:** Gbeti Collins, Peter Dzomeku, Michael Head.

**Writing – original draft:** Abdul-Wahab Inusah, Gbeti Collins, Peter Dzomeku, Michael Head, Shamsu-Deen Ziblim.

**Writing – review & editing:** Abdul-Wahab Inusah, Gbeti Collins, Peter Dzomeku, Michael Head, Shamsu-Deen Ziblim.

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
