## [Decision Letter · Decision Letter 0]

21 Oct 2022

PGPH-D-22-00970

Population knowledge, attitudes and practice towards Yellow Fever among nomadic populations: A cross-sectional study in Yellow Fever outbreak Communities in Ghana.

Dear Dr. Head,

Thank you for submitting your manuscript to PLOS Global Public Health. After careful consideration, we feel that it has merit but does not fully meet PLOS Global Public Health’s publication criteria as it currently stands. Therefore, we invite you to submit a revised version of the manuscript that addresses the points raised during the review process.

A response letter that responds to each point raised by the editor and reviewer(s). You should upload this letter as a separate file labeled 'Response to Reviewers'.A marked-up copy of your manuscript that highlights changes made to the original version. You should upload this as a separate file labeled 'Revised Manuscript with Track Changes'.An unmarked version of your revised paper without tracked changes. You should upload this as a separate file labeled 'Manuscript'.

We look forward to receiving your revised manuscript.

Kind regards,

Sarah Brewer, PhD

Academic Editor

Journal Requirements:

1. Please provide your detailed Financial Disclosure statement. This is published with the article. It must therefore be completed in full sentences and contain the exact wording you wish to be published.

a. Please clarify all sources of funding (financial or material support) for your study. List the grants (with grant number) or organizations (with url) that supported your study, including funding received from your institution. 

b. State the initials, alongside each funding source, of each author to receive each grant.

c. State what role the funders took in the study. If the funders had no role in your study, please state: “The funders had no role in study design, data collection and analysis, decision to publish, or preparation of the manuscript.”

d. If any authors received a salary from any of your funders, please state which authors and which funders.

If you did not receive any funding for this study, please simply state: “The authors received no specific funding for this work.

2. Please provide separate figure files in .tif or .eps format only and remove any figures embedded in your manuscript file. Please also ensure that all files are under our size limit of 10MB.

3. Figure 1: please (a) provide a direct link to the base layer of the map (i.e., the country or region border shape) and ensure this is also included in the figure legend; and (b) provide a link to the terms of use / license information for the base layer image or shapefile. We cannot publish proprietary or copyrighted maps (e.g. Google Maps, Mapquest) and the terms of use for your map base layer must be compatible with our CC-BY 4.0 license. 

Additional Editor Comments (if provided):

Reviewer 3 has raised concerns about the rigor of the statistical analysis and reporting as well as the use of standard English. Please pay special attention to these comments in your revisions. Additionally, the authors should review the appropriate reporting guidelines (https://journals.plos.org/globalpublichealth/s/submission-guidelines#loc-guidelines-for-specific-study-types) and criteria for publication (https://journals.plos.org/globalpublichealth/s/criteria-for-publication) to ensure they have included all the appropriate information about they study before resubmission.

The authors may also note that Reviewer 2 primarily focused on specific published works in this area and the authors may use their own judgement about the appropriateness of these suggestions for their manuscript.

Reviewers' comments:

Reviewer's Responses to Questions

**Comments to the Author**

1. Does this manuscript meet PLOS Global Public Health’s publication criteria? Is the manuscript technically sound, and do the data support the conclusions? The manuscript must describe methodologically and ethically rigorous research with conclusions that are appropriately drawn based on the data presented.

Reviewer #1: Yes

Reviewer #2: Yes

Reviewer #3: Partly

2. Has the statistical analysis been performed appropriately and rigorously?

Reviewer #1: Yes

Reviewer #2: Yes

Reviewer #3: No

3. Have the authors made all data underlying the findings in their manuscript fully available (please refer to the Data Availability Statement at the start of the manuscript PDF file)?

Reviewer #1: Yes

Reviewer #2: Yes

Reviewer #3: Yes

4. Is the manuscript presented in an intelligible fashion and written in standard English?

Reviewer #1: Yes

Reviewer #2: Yes

Reviewer #3: No

5. Review Comments to the Author

Reviewer #1: Title: Remove the first Population since there is another term population in the title. New title: Knowledge, attitudes, and practice towards Yellow Fever among nomadic

populations: A cross-sectional study in Yellow Fever outbreak Communities in Ghana.

**Figures**

Figure 1": The quality is bad. I suggest for the authors to submit a high resolution so that the labels are readable. Please include a map scale. Indicate the sampling sites or households as points or dots. It is also best to provide administrative boundaries if needed.

Page 11: Data Analsis: The data collected "from the _____" was exported to Microsoft Excel 2019 for cleaning.

Page 11: Data Analysis: For the scores of each dependent variables, I suggest to create a table.

Page 11 and 12: On the other hand, the key independent variables are __________.

Reviewer #2: I have the following comments for the authors to address and I am happy to review this paper again

1) Under the discussion, it would be good if the authors can compare and contrast their findings on Knowledge, attitude and practice (KAP) between yellow fever and other diseases such as dengue fever based on the following study:

Knowledge, Attitude and Practice about Dengue Fever among Patients Experiencing the 2017 Outbreak in Vietnam. Int J Environ Res Public Health. 2019 Mar 18;16(6):976. doi: 10.3390/ijerph16060976. PMID: 30889912; PMCID: PMC6466316.

2) Participants from Ghana from stay in rural areas and from lower social class. Please discuss strategies from other infectious studies (e.g. COVID pandemic) to improve KAP based on the methods proposed by following studies and include in the discussion:

Strengthen health care system in resource scarce setting:

Strengthening Health System and Community Responses to Confront COVID-19 Pandemic in Resource-Scare Settings. Front Public Health. 2022 Jul 6;10:935490. doi: 10.3389/fpubh.2022.935490. PMID: 35875028; PMCID: PMC9296814.

Involvement grassroot system:

Feasibility of Intersectoral Collaboration in Epidemic Preparedness and Response at Grassroots Levels in the Threat of COVID-19 Pandemic in Vietnam. Front Public Health. 2020 Nov 17;8:589437. doi: 10.3389/fpubh.2020.589437. PMID: 33313040; PMCID: PMC7707108.

Involvement of village health collaborators:

Reaching further by Village Health Collaborators: The informal health taskforce of Vietnam for COVID-19 responses. J Glob Health. 2020;10(1):010354. doi:10.7189/jogh.10.010354

Reviewer #3: This manuscript describes a cross-sectional survey to evaluate the knowledge, attitude and practices of nomadic populations in Ghana about towards Yellow Fever. The results showed poor level of knowledge about causes, transmission, signs and symptoms of the disease; over 80% and 72% of the participants had positive attitude and positive practices, respectively. Significant predictors of positive attitude and practices were identified.

Overall, the study is interesting and important; however, the manuscript has major drawbacks that should be addressed before it can be acceptable for publication. These include the quality of writing, data analysis and presentation of results and discussion of findings. Specific comments are listed below.

MAJOR REVISION

1. The quality of writing should be improved. The manuscript needs extensive stylistic and English editing to correct several grammatical and typographical errors. For instance, capitalized letters in 1st line in abstract: Global; 4th paragraph of abstract: Age, Nationality; duration of in (stay is missing); 5th paragraph of abstract: Nationality, Positive; Introduction: jaundice body pain, all subsections titles and several variables in all tables, etc. Some other mistakes are indicated in minor revision.

2. Statistical analysis and results reporting and presentation should also be improved. For instance:

2.1. Reporting of results in abstract should be improved. Add statistics for significant factors (predictors) associated with positive attitude and positive practices, where applicable.

2.2. Why predictors of poor or good knowledge were not identified?

2.3. Table 4: Why all variables were included in the multivariate analysis? This should be explained.

2.4. Table 5: Why all variables were included in the multivariate analysis? This should be explained.

3. Discussion section is too lengthy and should be entirely rewritten. It should be shortened substantially focusing on important findings and the important message to policy-makers. Over repetition of results should be avoided.

4. The conclusion (in abstract and main section) is not supported by the findings. Yes, the population had poor knowledge about yellow fever, and this needs proper intervention; however, the impact of any intervention like health education is still not evaluated. Main conclusion should be improved.

MINOR REVISION (arranged following text flow)

ABSTRACT:

5. Results in the 4th paragraph and 5th paragraph: it should be indicated if these results were obtained by univariate or multivariate analyses. Significant associations or not? Add statistics.

6. 5th paragraph: what was the most commonly used strategy?

7. Last paragraph: “….. yellow fever epidemic control can be improved in hard-to-reach communities through locally-tailored education and health promotion campaigns …” The conclusion is not supported by the findings. Yes, the population had poor knowledge about yellow fever, and this needs proper intervention; however, the impact of any intervention like health education is still not evaluated.

INTRODUCTION:

8. 3rd sentence: rephrase to improve clarity.

9. “The virus”: which virus?

10. 2nd paragraph: “in the continent”. Which continent?

11. 6th paragraph: “In Iran, ….”. This paragraph should be rewritten to provide better justification of the study.

12. Information on the control program in Ghana should be provided?

METHODS:

13. Study Area, Design and Period: Areas’ epidemiological features related to vector-borne infections can be added.

14. Study Population, Participants, and inclusion Criteria: 2nd paragraph “Using the yellow fever line list ……. were purposively selected for the study.” This should be moved to previous section (study area). Information on invitations to participate, acceptance or rejection and related numbers should be added.

15. “Using snowballing approaches,”: does this mean that other population groups resided in the targeted areas?

16. “A total population of 403 nomadic households”: please rephrase.

17. Key dependent variables were; knowledge, attitudes and practices towards yellow fever. This should be merged with related paragraph.

18. “were considered as having “Positive knowledge”,” can be changed to “Good knowledge”.

19. This scoring system is adapted from previous similar studies. (13). But only a single study was cited.

20. 24 variables; 12 variables, using 9 variables. These can be items or questions.

21. 60% cut-off point: How the 60% was calculated? The scores were either 1 or 0. Explain.

RESULTS:

22. Table 1: define HH Size in footnote. Religious affiliation can be removed and indicated in the text.

23. In this study, only those who have heard of yellow fever were included the study. This is a vague statement. Does it mean that all the 414 participants heard about yellow fever? Did you ask at the beginning and only recruit those who had prior knowledge? This should be clearly explained.

24. “… a bite of an infected mosquito” or a bite of mosquito?

25. About 47% believed yellow can be … Another example of writing mistakes.

26. Table 2: Monkey transmit YF to person. Revise!

27. YF can breed inside homes. Revise!

28. Overall knowledge was stated in a mean score. Why this was not following attitude and practice method?

29. Nomadic Attitudes towards Yellow Fever: Some of the questions are related to knowledge and not to attitude.

30. Table 3 (last variable) shows that 80.2% had negative attitude. This contradicts the results in previous text and in abstract “Over 80% of household heads surveyed had positive attitudes”.

31. Overall Attitudes toward yellow fever: Table 4. But it is displayed in Table 3. Does this analysis include all participants or only those with

32. Table 4: (n) of each category must be added.

33. Table 5: (n) of each category must be added.

34. Why males had lower level of correct or positive practices compared to females, although they have significantly higher level of positive attitude?

DISCUSSION:

35. Comparisons with previous findings should be justified and explained. For example, reference #13 cited in the 3rd line can be invalid as the study was among university students and in Ethiopia.

36. “This affirms the fact that yellow fever is endemic in Ghana (3).” This and subsequent related sentences can be removed.

37. Conclusion: It is important to ensure that there is good population knowledge, attitude and practice …. Can be removed. What are the important findings? What are the implications? What are your recommendations?

38. Figure 1: The current figure is not useful. The study areas should be indicated on the map. Regions’ boundaries should be displayed.

39. Reference list should be prepared following the journal’s instructions and style.

6. PLOS authors have the option to publish the peer review history of their article (what does this mean?). If published, this will include your full peer review and any attached files.

**Do you want your identity to be public for this peer review?** For information about this choice, including consent withdrawal, please see our Privacy Policy.

Reviewer #1: **Yes: **Thaddeus M. Carvajal

Reviewer #2: No

Reviewer #3: **Yes: **Hesham M. Al-Mekhlafi

---

## [Decision Letter · Decision Letter 1]

15 Feb 2023

Knowledge, attitudes and practice towards Yellow Fever among nomadic populations: A cross-sectional study in Yellow Fever outbreak Communities in Ghana

PGPH-D-22-00970R1

Dear Dr Head,

We are pleased to inform you that your manuscript 'Knowledge, attitudes and practice towards Yellow Fever among nomadic populations: A cross-sectional study in Yellow Fever outbreak Communities in Ghana' has been provisionally accepted for publication in PLOS Global Public Health.

Best regards,

Julia Robinson

Executive Editor

Reviewer Comments (if any, and for reference):

Reviewer's Responses to Questions

**Comments to the Author**

1. If the authors have adequately addressed your comments raised in a previous round of review and you feel that this manuscript is now acceptable for publication, you may indicate that here to bypass the “Comments to the Author” section, enter your conflict of interest statement in the “Confidential to Editor” section, and submit your "Accept" recommendation.

Reviewer #2: All comments have been addressed

2. Does this manuscript meet PLOS Global Public Health’s publication criteria? Is the manuscript technically sound, and do the data support the conclusions? The manuscript must describe methodologically and ethically rigorous research with conclusions that are appropriately drawn based on the data presented.

Reviewer #2: Yes

3. Has the statistical analysis been performed appropriately and rigorously?

Reviewer #2: Yes

4. Have the authors made all data underlying the findings in their manuscript fully available (please refer to the Data Availability Statement at the start of the manuscript PDF file)?

Reviewer #2: Yes

5. Is the manuscript presented in an intelligible fashion and written in standard English?

Reviewer #2: Yes

6. Review Comments to the Author

Reviewer #2: I recommend publication

7. PLOS authors have the option to publish the peer review history of their article (what does this mean?). If published, this will include your full peer review and any attached files.

**Do you want your identity to be public for this peer review?** For information about this choice, including consent withdrawal, please see our Privacy Policy.

Reviewer #2: No
